# Modulation of Viral Programmed Ribosomal Frameshifting and Stop Codon Readthrough by the Host Restriction Factor Shiftless

**DOI:** 10.3390/v13071230

**Published:** 2021-06-25

**Authors:** Sawsan Napthine, Chris H. Hill, Holly C. M. Nugent, Ian Brierley

**Affiliations:** Department of Pathology, University of Cambridge, Cambridge CB2 1QP, UK; sn459@cam.ac.uk (S.N.); chh30@cam.ac.uk (C.H.H.); hcmn2@cam.ac.uk (H.C.M.N.)

**Keywords:** ribosome, frameshift, readthrough, Shiftless, virus

## Abstract

The product of the interferon-stimulated gene *C19orf66*, Shiftless (SHFL)*,* restricts human immunodeficiency virus replication through downregulation of the efficiency of the viral *gag/pol* frameshifting signal. In this study, we demonstrate that bacterially expressed, purified SHFL can decrease the efficiency of programmed ribosomal frameshifting in vitro at a variety of sites, including the RNA pseudoknot-dependent signals of the coronaviruses IBV, SARS-CoV and SARS-CoV-2, and the protein-dependent stimulators of the cardioviruses EMCV and TMEV. SHFL also reduced the efficiency of stop-codon readthrough at the murine leukemia virus *gag/pol* signal. Using size-exclusion chromatography, we confirm the binding of the purified protein to mammalian ribosomes in vitro. Finally, through electrophoretic mobility shift assays and mutational analysis, we show that expressed SHFL has strong RNA binding activity that is necessary for full activity in the inhibition of frameshifting, but shows no clear specificity for stimulatory RNA structures.

## 1. Introduction

Recoding events during mRNA translation have been described in the expression of many viral and cellular genes and include such phenomena as programmed ribosomal frameshifting (PRF), stop codon readthrough (RT), selenocysteine insertion and translational bypassing [1,2]. In –1 PRF, mRNA signals induce ribosomes to slip back by one nucleotide (nt) into an overlapping open reading frame (ORF) and to continue translation, generating a fusion protein of upstream and downstream ORFs [3,4,5,6,7,8,9,10]. First described as the mechanism by which the Gag-Pol polyprotein of the retrovirus Rous sarcoma virus (RSV) is expressed from overlapping *gag* and *pol* ORFs [11], related –1 PRF signals have been documented in many other viruses of clinical, veterinary and agricultural importance [1,5]. PRF has also been increasingly recognized in cellular genes of both prokaryotes and eukaryotes as well as in other replicating elements, such as insertion sequences and transposons [12,13,14]. –1 PRF is associated with diverse biological processes, including the expression of viral replicases (e.g., HIV reverse transcriptase [15]), bacteriophage tail fibers [16] and the Escherichia coli (*E. coli*) DNA polymerase γ subunit [17]. It is also involved in the regulation of mRNA stability [18,19], the cell cycle [14], mammalian development [20], human paraneoplastic disorders [21] and genetic diseases [22].

Central to –1 PRF is the interaction of the ribosome with a stimulatory mRNA structure (a stem-loop or pseudoknot) that promotes frameshifting at the slippery sequence (Figure 1). How these RNA structures act is incompletely understood, but by presenting an unusual topology [3,4,23,24,25], they likely confound an intrinsic unwinding activity of the ribosome with consequent effects on the elongation cycle and frame maintenance [26,27,28]. Indeed, kinetic analyses in bacterial systems indicate that stimulatory RNAs can impair movements of the ribosomal small subunit (30S) head, delaying the dissociation of EF-G, and the release of tRNA from the ribosome [29,30,31]. The propensity of the stimulatory RNA to sample multiple folded conformations has also been linked to –1 PRF efficiency [32,33,34,35] and an effect of co-translational folding of the nascent peptide can also modulate the process [36]. Mechanistic insights into –1 PRF are also being sought through studies of two highly efficient PRF signals whose activity requires the participation of *trans*-acting proteins (Figure 1). In the cardioviruses encephalomyocarditis virus (EMCV) and Theiler’s murine encephalitis virus (TMEV), frameshifting within the 2B gene requires viral protein 2A [37,38,39], and in the arterivirus porcine reproductive and respiratory syndrome virus (PRRSV), production of the nsp2TF protein by frameshifting is mediated by a complex of the viral protein nsp1β and host poly[C] binding protein (PCPB) [40,41,42].

Recently, an interferon (IFN)-stimulated gene (ISG) was identified whose encoded product, Shiftless (SHFL)*,* is able to restrict human immunodeficiency virus (HIV) replication through *downregulation* of the efficiency of the viral *gag/pol* frameshifting signal [43]. SHFL is the first identified cellular factor that acts as a repressor of –1 PRF and the first example of an ISG that targets a translational recoding process. It is a 33 kDa protein encoded in an eight-exon gene (*C19orf66*) located on genomic region 19p13.2. The protein has also been referred to as C19orf66 in studies of its role in restricting Kaposi’s sarcoma-associated herpes virus (KSHV) [44], hepatitis C virus (HCV) [45] and Zika virus (ZIKV) [46]; by virtue of its ability to repress the yield of Dengue virus (DENV), it has also been named RyDEN [47] and interferon-regulated antiviral gene (IRAV) [48]. In virus replication and –1 PRF reporter gene assays in cells, SHFL was found to be a broad-spectrum inhibitor of –1 PRF [43] but did not greatly affect replication of the retrovirus murine leukemia virus (MuLV), suggesting that the protein may not inhibit pseudoknot-dependent RT, a related recoding process employed in the expression of the MuLV Gag-Pol polyprotein [49]. The extent of inhibition of –1 PRF found in these experiments was typically 50–60%, a magnitude of reduction known to suppress replication of a variety of viruses employing PRF [50,51,52,53,54]. Through polysome profiling, SHFL was found to associate with actively translating ribosomes, and co-immunoprecipitated with proteins from both the small (eS31) and large (uL5) subunits in a binding screen of individual ribosomal proteins. SHFL was also able to bind to a short RNA containing the HIV-1 PRF signal, although the affinity and specificity of the interaction were not reported.

In this paper, we demonstrate that bacterially expressed, purified SHFL can reduce the efficiency of both PRF and RT in in vitro translation assays and confirm binding of the protein to mammalian ribosomes in vitro. Through electrophoretic mobility shift assays and mutational analysis, we show that the expressed protein has the strong RNA binding activity necessary for full activity in the inhibition of recoding but shows no obvious specificity for stimulatory RNA structures.

## 2. Materials and Methods

### 2.1. Plasmids

Assessment of in vitro recoding efficiencies employed the reporter plasmids p2luc [55] and pDluc [56], with viral sequences inserted between the *Renilla* and firefly luciferase coding sequences as detailed in Table 1. Plasmid pHis-SHFL was cloned as follows. The human SHFL cDNA sequence was codon-optimized for *E. coli* expression and synthesized de novo (Epoch Life Sciences, Sugar Land, TX, USA). This was subsequently amplified by PCR (F 5′ AATTCATATGTCTCAGGAAGGTGTGGA 3′; R 5′ AATTGGATCCTTATTACTCGCGGGGCCCGCCCTCCT 3′) and cloned into *Nde* I/*Bam*H 1-cut pOPT3H to introduce an N-terminal His_6_ tag.

### 2.2. Protein Expression and Purification

Recombinant protein was produced in *E. coli* BL21 (DE3) pLysS cells grown by shaking (210 rpm) in 2×TY broth supplemented with 100 µg/mL ampicillin and 12.5 µg/mL chloramphenicol (37 °C). Expression was induced at A_600_ of ~0.6 with 0.5 mM isopropyl β-D-1-thiogalactopyranoside (IPTG; 21 °C, 16 h). Cells were harvested by centrifugation (4000× *g*, 4 °C, 20 min), washed in ice-cold phosphate-buffered saline (PBS) and stored at −20 °C. Protein was prepared from 6L culture. Per liter, cell pellets were resuspended in 50 mL ice-cold lysis buffer (50 mM Tris (HCl) pH 7.9, 500 mM NaCl, 30 mM imidazole, 0.05% *w/v* Tween-20 supplemented with 50 μg/mL DNase I and EDTA-free protease inhibitors) and lysed using a cell disruptor (Constant Systems, Daventry, UK; 24 kPSI, 4 °C). The lysate was clarified by centrifugation (39,000× *g*, 30 min, 4 °C) and incubated (1 h, 4 °C) with 0.5 mL Ni-NTA agarose beads (Qiagen, UK) that had been pre-equilibrated in the same buffer. Beads were washed three times by centrifugation (600× *g*, 10 min, 4 °C) and resuspension in 50 mL wash buffer (50 mM Tris (HCl) pH 7.9, 500 mM NaCl, 30 mM imidazole). Washed beads were pooled, transferred to a gravity column, and protein was eluted dropwise with 50 mM Tris (HCl) pH 7.9, 300 mM NaCl, and 300 mM imidazole. Fractions containing SHFL were pooled and dialyzed into 20 mM Tris (HCl) pH 7.9, 300 mM NaCl, 1.0 mM DTT (10 K molecular weight cut-off (MWCO), 4 °C, 16 h). To remove nucleic acids, dialysate was loaded onto a 5 mL HiTrap Heparin column (GE Healthcare, UK) at 2.0 mL/min, washed with two column volumes buffer (25 mM Tris (HCl) pH 7.9, 300 mM NaCl, 1.0 mM DTT) and eluted with a 0–100% gradient of high-salt buffer (25 mM Tris (HCl) pH 7.9, 1.0 M NaCl, 1.0 mM DTT) over 20 column volumes. SHFL-containing fractions were pooled and concentrated using an Amicon^®^ Ultra centrifugal filter unit (10K MWCO, 4000× *g*) prior to further purification by size exclusion chromatography (Superdex 200 10/300 column; 25 mM Tris (HCl) pH 7.9, 300 mM NaCl, 1.0 mM DTT). Purity was assessed by 4–20% gradient SDS-PAGE, and protein identity was verified by mass spectrometry. Purified protein was concentrated (~6.5 mg/mL, 187 μM), snap-frozen in liquid nitrogen and stored at −80 °C. 

### 2.3. In Vitro Translation

Frameshift reporter plasmids were linearized with *Eco*R I (p2luc) or *Fsp* I (pDluc), and capped run-off transcripts generated using T7 RNA polymerase. Messenger RNAs were translated in nuclease-treated rabbit reticulocyte lysate (RRL) extracts (Promega) programmed with ~50µg/mL template mRNA. Typical reactions were of 10 µL volume and composed of 90% (*v/v*) RRL, 20 µM amino acids (lacking methionine) and 0.2 MBq [^35^S]-methionine. When added, SHFL was diluted where necessary in dilution buffer (DB) (5 mM Tris (HCl) pH 7.5, 100 mM KCl, 1 mM DTT, 0.05 mM EDTA, 5% glycerol) and added to reactions in 1 µL final volume. Reactions were incubated for 1 h at 30 °C and stopped by the addition of an equal volume of 10 mM EDTA, 100 µg/mL RNase A, followed by incubation at room temperature for 20 min. Samples were prepared for SDS-PAGE by the addition of 10 volumes of 2× sample buffer, boiled for 4 min and resolved on 12% gels. Dried gels were exposed to a Cyclone Plus Storage Phosphor Screen (PerkinElmer, Waltham, MA, USA), the screen scanned using a Typhoon TRIO Variable Mode Imager (GE Healthcare, UK) in storage phosphor autoradiography mode, and bands were quantified using ImageQuant^TM^TL software (GE Healthcare, UK). The calculations of frameshifting efficiency (% –1FS) took into account the differential methionine content of the two products and % –1FS was calculated as = 100 × (IFS/MetFS)/(IS/MetS + IFS/MetFS). In the formula, the number of methionines in the stop and –1 FS products are denoted by MetS and MetFS; while the densitometry values for the same products are denoted by IS and IFS, respectively. Calculations of readthrough efficiency followed the same procedure.

### 2.4. Immunoblotting

Aliquots of fractions from size exclusion chromatography were denatured in an equal volume of 2X sample buffer and heated to 95 °C for 5 min. Proteins were separated on 10–20% SDS-PAGE gradient gels (Novex; Thermo Fisher Scientific, Waltham, MA, USA) and transferred to nitrocellulose membranes. These were blocked for 60 min with 5% powdered milk (Marvel) in PBS + 0.1% Tween-20 (PBST) and probed at 4 °C overnight with primary antibodies (mouse anti-RPL4 (clone 4A3, Merck, UK, WH0006124M1 at 1:1000); mouse anti-RPS6 (clone A16009C, BioLegend, UK, 691802 at 1:1000); rabbit anti-SHFL (HPA042001, Atlas Antibodies at 1:500)). Membranes were washed in PBST before being incubated in the dark with an IRDye-conjugated secondary antibody in PBST. Blots were scanned using an Odyssey infrared imaging system (Li-Cor).

### 2.5. Electrophoretic Mobility Shift Assay (EMSA)

Short, ^32^P-labelled template RNAs (48-102 nt) containing individual recoding signals were prepared by T7 transcription of linearized plasmid templates. SHFL dilutions were in dilution buffer (DB) (5 mM Tris (HCl) pH 7.5, 100 mM KCl, 1 mM DTT, 0.05 mM EDTA, 5% glycerol) and added to reactions (10 µL final volume) alongside EMSA buffer (10 mM HEPES (NaOH) pH 7.6, 150 mM KCl, 2 mM MgCl_2_, 1 mM DTT, 0.5 mM ATP, 5% glycerol, 100 µg/mL porcine tRNA, 10U RNase inhibitor per mL), after which the radiolabeled probe was introduced. After incubation at 37 °C for 15 min, samples were loaded promptly onto 10% acrylamide non-denaturing gels (acrylamide:bisacrylamide ratio 10:1) and run at 175 V at room temperature until free and bound RNA species were resolved. Gels were fixed for 15 min in 10% acetic acid, 10% ethanol, dried and exposed to X-ray film.

## 3. Results

### 3.1. Effect of Purified SHFL on Ribosomal Frameshifting and Readthrough In Vitro

Wang and colleagues [43] studied the involvement of SHFL in –1 PRF in tissue culture cells by measuring the levels of relevant virus proteins in infected cells expressing SHFL, or by transfection of frameshift reporter plasmids. We wished to determine if the purified protein had anti-frameshifting activity in vitro. Codon-optimized, N-terminally-His_6_-tagged SHFL was expressed in *E. coli* BL21 (DE3) pLysS, purified and used to supplement in vitro translation reactions (rabbit reticulocyte lysate (RRL)) programmed with frameshift-reporter mRNAs (see Materials and Methods). SDS-PAGE of the purified protein confirmed its integrity (Figure 2), and its identity was confirmed by mass spectroscopy (Mr = 33,109 Da, 60% peptide coverage). The published capacity to bind 80S ribosomes [43] was verified by size-exclusion chromatography (Figure 2B,C). In in vitro translation experiments, we employed the HIV-1 signal as a positive control (following [43]), and an assortment of other –1 PRF signals, including the RNA pseudoknot-dependent signals of the betacoronavirus SARS-CoV and the gammacoronavirus infectious bronchitis virus (IBV), and the protein-dependent signals of EMCV and TMEV (Figure 3). At all signals tested, SHFL addition led to a reduction in –1 PRF, varying from a modest effect at the HIV *gag/pol* signal (reduced from 4.5 to ~3%; Figure 3) to a more substantial reduction at the protein-dependent site of TMEV (from 53% to 12%). A version of SHFL with reduced RNA binding capacity (mutant M1, see below) was also expressed and tested in comparison to the wild-type protein at the TMEV signal (Figure 3). As can be seen, the addition of the mutant protein had little effect on –1 PRF. We also tested the capacity of SHFL to modulate stop codon readthrough, using the signal derived from the *gag/pol* overlap of the retrovirus murine leukemia virus (MuLV). We found that SHFL was indeed able to decrease readthrough efficiency, four-fold in RRL (Figure 3).

### 3.2. RNA Binding Activity of SHFL

How SHFL affects recoding processes is not well understood but may involve specific recognition of the stimulatory RNAs. To assess RNA binding, we carried out electrophoretic mobility shift assays (EMSAs) using 5′-end-labeled transcripts (48-102 nt in length) containing the frameshift-stimulatory RNA of IBV (both the minimal PK (IBVmin) with shortened L3 [58] and the wild-type pseudoknot (IBVmax)), the MuLV RT signal [54] or a variant destabilized in PK stem 1 (MuLV S1) and a control transcript derived from the SARS-CoV-2 antigenome with no obvious recoding signal (control). As seen in Figure 4, protein-RNA complexes were formed in all cases at similar SHFL concentrations, except for IBVmin, where the binding was noticeably weaker. Increasing SHFL concentration generally led to a concomitant stepwise reduction in mobility of the SHFL-RNA complexes, although the fastest mobility complex persisted somewhat with the MuLV RNA. Overall, these data indicate that SHFL has strong RNA binding activity but appears to lack obvious specificity to sites of –1 PRF, and probably, RT. Computational analysis of the SHFL protein sequence reveals a number of putative functional domains within a predicted secondary structure comprising 13 α-helixes and four β-strands (Figure 5; [47]). Using PHYRE2 [59], we identified several arginine residues within a putative zinc-finger domain that are potentially involved in nucleic acid binding. We prepared three derivatives of SHFL with alanine substitutions at various positions (mutant 1, R131A/R136A; mutant 2, R131A/R133A; and mutant 3, R131A/R133A/R136A), expressed and purified the proteins, and tested their capacity to inhibit SARS-CoV-2 PRF in RRL and to bind RNA in EMSAs using the IBVmax template (Figure 6; Appendix A). We found a correlation between RNA binding and the inhibition of PRF, with the wild-type protein most active in each case and M3 the least active. These experiments indicate that RNA binding is required for full activity in the inhibition of PRF. 

## 4. Discussion

In this study, we show that recombinant SHFL can decrease the efficiency of programmed ribosomal frameshifting in vitro at a variety of sites of viral –1 PRF and can also have an impact on programmed stop-codon readthrough at the MuLV signal. We demonstrate that the expressed protein has RNA binding activity, identify arginine residues that contribute to this, and show that RNA binding is necessary for full activity in the inhibition of recoding.

In recoding assays in RRL, the effect of SHFL varied, with the recombinant protein invoking reductions in the region of 0.5 to four-fold. We noted a general inhibitory effect of SHFL on overall translation at higher levels of SHFL, a possible consequence of its ribosome binding capacity, which could compromise elongation. This inhibitory effect was observed only at concentrations of SHFL higher than those having an effect on recoding, however, indicating the specificity of SHFL action in inhibiting recoding events (densitometry of the protein bands of Figure 3 is shown in the Appendix A). Nevertheless, subtle effects on elongation, coupled with minor differences between related recoding signals and their reporter mRNA contexts, might account for the observed differences in the magnitude of the SHFL effect. In addition, the relative levels of potential co-factors in RRL extracts might influence the activity of SHFL at some sites. The effect of SHFL at the HIV-1 signal was modest, but in line with that measured in transfected cells, with frameshifting at the HIV-1 and SARS-CoV-2 signals known to be reduced to some 60–70% of wild-type levels in HEK293 cells [62].

In EMSAs, SHFL bound a variety of short RNA templates (apparent Kd~500 nM) with no obvious specificity for templates containing a functional stimulatory RNA. In these experiments, there was a trend towards decreased mobility of the RNA-protein complex as the relative concentration of SHFL was increased, potentially signifying the generation of higher-order complexes with multiple copies of SHFL on the RNA. Alternatively, SHFL has been proposed to self-associate via interactions between putative C-terminal coiled-coil domains [43], therefore, some higher-order complexes may involve dimeric species. In the MuLV EMSA, the slower mobility complexes appeared alongside a more persistent band that may be a 1:1 complex. Further work will be required to ascertain whether this reflects a specific binding event. The observed impaired binding to the minimal IBV pseudoknot (Figure 4) can be accounted for by the compact nature of the pseudoknot on this template, which lacks extensive regions of single-stranded RNA (this variant of the IBV stimulatory pseudoknot has a much shorter loop 3, reduced from 32 to 8 nt [58]). These data are consistent with the studies of Balinsky and colleagues [48], where SHFL was shown to bind more strongly to single-stranded RNA than to double-stranded DNA. SHFL variants with mutations in arginine residues potentially involved in nucleic acid binding showed reduced activity in both RNA binding and inhibition of PRF. Further characterization of these mutants (M1, M2, M3) will be required to understand the molecular basis of the deficit. We note, for example, that the M1 mutant had no obvious activity against the protein-dependent TMEV signal (Figure 3) but was able to impair SARS-CoV-2 frameshifting, albeit at higher concentrations than the wild-type protein (Figure 6). This may reflect mechanistic differences in –1 PRF between the two signals.

How SHFL downregulates –1 PRF is uncertain. Wang and colleagues [43] have suggested that the protein binds to ribosomes when they are in a non-canonical rotated state (the postulated frameshifting state) [29,30,31,63], leading to ribosome pausing. Subsequently, stalled ribosomes are thought to be removed from the template in a process involving recruitment of the release factor complex (eRF1-eRF3) leading to the generation of a premature termination product (Figure 1). The data described in the present work are supportive of this model, with SHFL acting most likely through an interaction with ribosomes rather than via specific binding to the stimulatory RNAs. A capacity to bind RNA is essential for maximal activity, but whether this reflects a necessity to bind to rRNA or mRNA, or both, remains to be determined. The role of release factors in the pathway of SHFL action is also uncertain. In the present study, we cannot easily discriminate proteins derived from natural termination at the upstream cistron of the reporter mRNA (the Stop product on the gels) from any that may arise from drop-off at the recoding site, as the sites are too close together on the mRNA. However, the observation that SHFL can reduce programmed RT in vitro is consistent with a role for SHFL in recruiting release factors. It has been shown previously that expressed SHFL was not found to have a substantial effect on MuLV replication in culture [43]. Based on our previous work [54], the ~four-fold reduction in readthrough we see in RRL might be expected to noticeably reduce MuLV replication in culture. However, differences in experimental systems might account for this discrepancy.

SHFL is an important antiviral—and likely broadly antipathogenic—ISG discovered relatively recently [47] and is, as yet, not extensively studied [43,44,45,46,47,48,64]. How it exerts its effect in different viral contexts is not well established. In Zika virus, SHFL degrades viral protein NS3 [46], whereas, in DENV, which also lacks a documented –1 PRF signal, it may act by interfering with viral mRNA translation through the formation of an inhibitory complex [47] with polyA binding protein C1 (PABPC1; known to bind to the DENV 3′ UTR [65]) and La autoantigen motif-related protein 1 (LARP1). Balinski and colleagues [48] alternatively hypothesize that DENV replication is inhibited through degradation of viral mRNA, through the recruitment of replication complexes to processing (P) bodies following interactions between SHFL and cellular factors including MOV10 (a RISC complex RNA helicase). In HCV, SHFL restricts the formation of the viral replication organelle, likely through interactions with stress granule and P-body components [45]. In KSHV, SHFL restricts early gene expression and particle release, perhaps also through interactions with PABPC1 and LARP1. Along with its role in inhibiting –1 PRF and RT, these studies reveal the capacity of SHFL to associate with a variety of cellular factors and to mediate its antiviral activity through different pathways. The activity of recombinant SHFL in reducing recoding efficiencies in RRL does not rule out an involvement of co-factors, as homologs of some known interacting partners [e.g., PABPC1] are present in this extract. Further work will be required to decipher the molecular mechanisms of SHFL action in virus-infected cells.

## Figures and Tables

**Figure 1 viruses-13-01230-f001:**
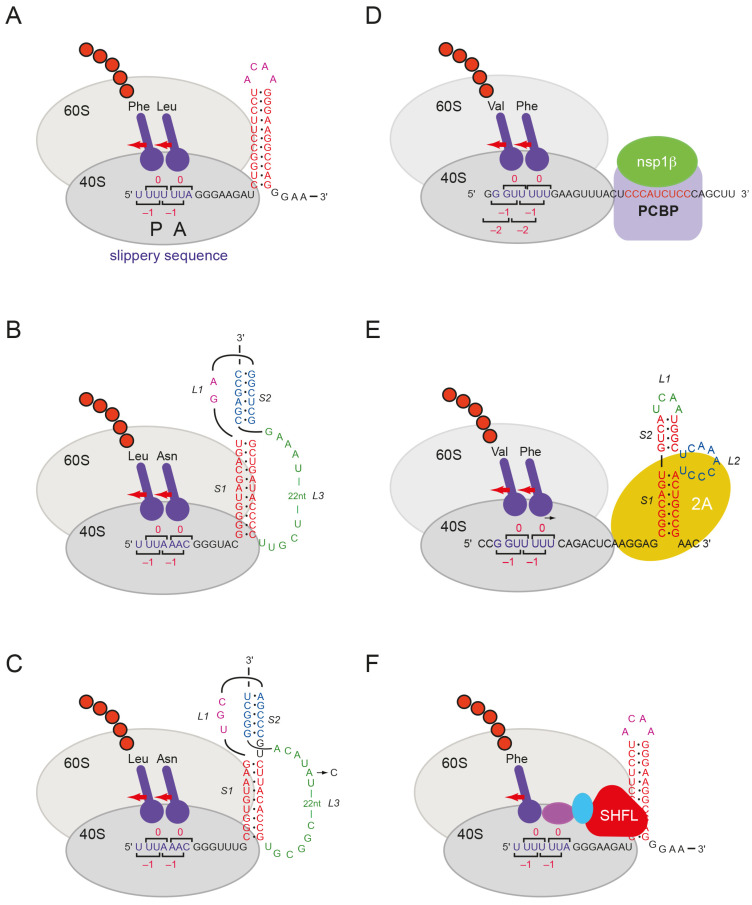
Modulators of programmed –1 ribosomal frameshifting. Cartoon drawings of ribosomes encountering sites of PRF. *Stimulators*: (**A**) HIV stem-loop; (**B**) IBV pseudoknot; (**C**) SARS-CoV-2 pseudoknot (SARS-CoV sequence identical in the region shown except for an (**A**) to (**C**) transversion in loop 3 (L3) as indicated); (**D**) PRRSV complex of virus (nsp1β) and host protein (PCBP); (**E**) Cardiovirus (EMCV) stimulatory RNA complexed with viral 2A protein. *Repressor*: (**F**) SHFL shown bound at the HIV-1 signal. Purple and cyan ovals represent eRF1 and eRF3 (in complex).

**Figure 2 viruses-13-01230-f002:**
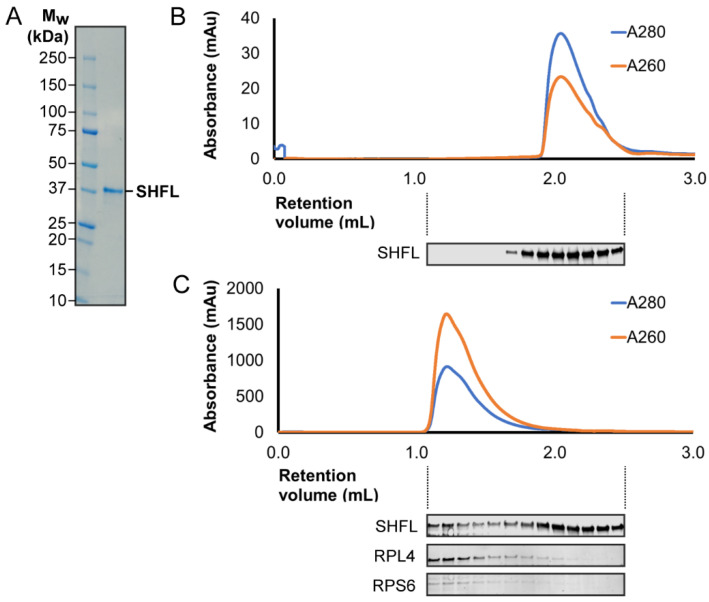
Characterization of purified SHFL and association with mammalian 80S ribosomes**.** (**A**) SDS-PAGE analysis of purified SHFL. (**B**) Size-exclusion chromatogram of purified SHFL (37 µM). Fractions across the peak were analyzed by immunoblot for SHFL. (**C**) As (**B**) following the incubation of SHFL (18 µM) with RRL 80S ribosomes (1 µM; purified according to [57]). Fractions across the peak were immunoblotted for SHFL and ribosomal proteins uL4 and eS6. Note, in panel (**C**), the absorbance readings are dominated by 80S ribosomes.

**Figure 3 viruses-13-01230-f003:**
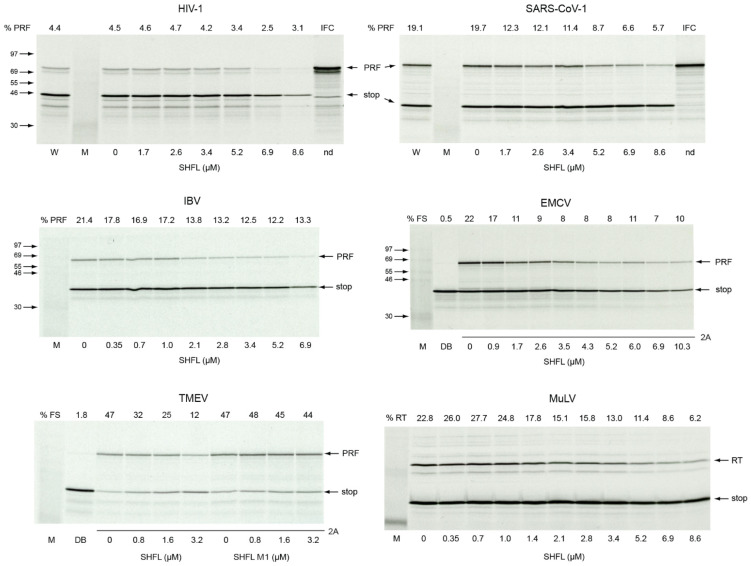
Inhibition by SHFL of PRF and RT in vitro**.** RNAs derived from recoding reporter plasmids were translated in RRL in the presence of the indicated concentration of His-tagged SHFL or SHFL-M1. The addition of only water (W) or SHFL dilution buffer (DB) is also indicated. The products were resolved by 12% SDS-PAGE and visualized by autoradiography. ^14^C molecular size markers were also run on the gel (M). IFC represents an in-frame control translation of an mRNA in which the two reading frames are placed in-frame by the addition of a single nucleotide immediately downstream of the slippery sequence (see Table 1). Proteins generated by ribosomes that do not frameshift (stop) or that enter the −1 reading frame (PRF), or read through a stop codon (RT) are arrowed. In the EMCV and TMEV panels, the addition of the *trans*-activator protein 2A is shown by lane underlining.

**Figure 4 viruses-13-01230-f004:**
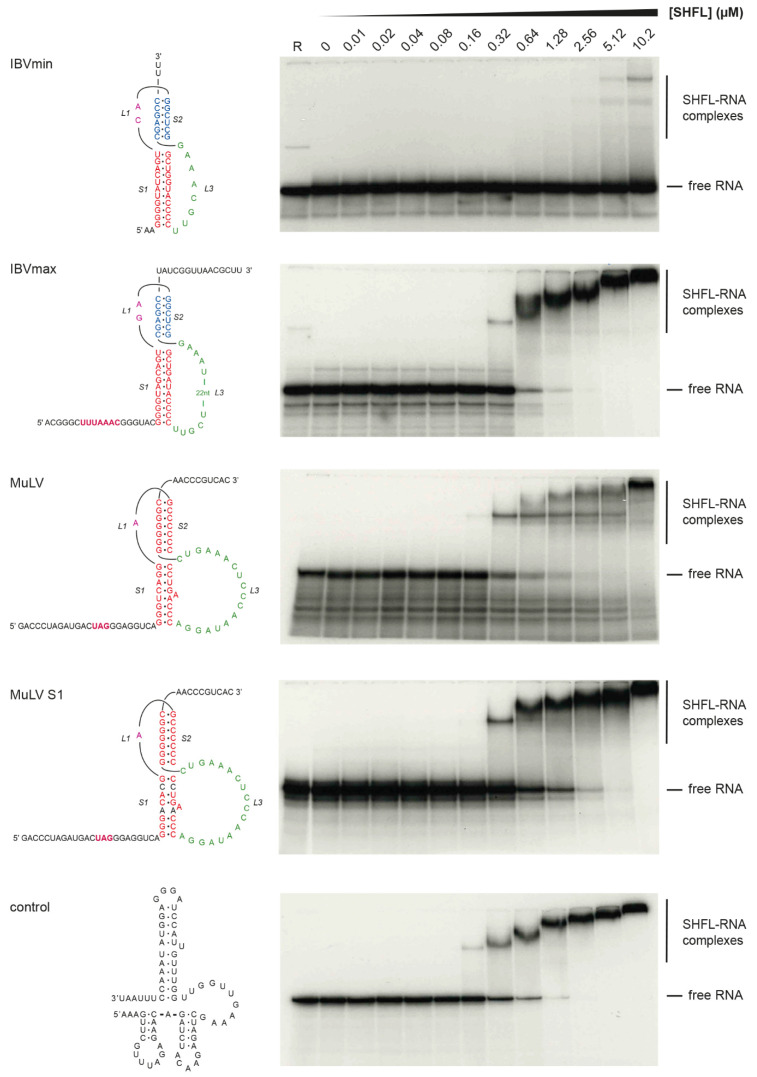
SHFL binds to RNA in EMSA assays. EMSA analysis of binding of SHFL to short (48-102 nt) ^32^P-labeled in vitro transcripts containing a recoding signal, or a control RNA (as indicated on the left). After incubation at 37 °C for 15 min, reactions were loaded onto 10% non-denaturing polyacrylamide gels and, following electrophoresis, the gels were fixed, dried and subjected to autoradiography. The radiolabeled RNA was at 10 nM in each reaction. R represents a lane loaded with RNA only. In lanes marked 0, the RNA was incubated alone with SHFL dilution buffer (see Materials and Methods). Destabilized base-pairs in MuLV S1 are in black.

**Figure 5 viruses-13-01230-f005:**
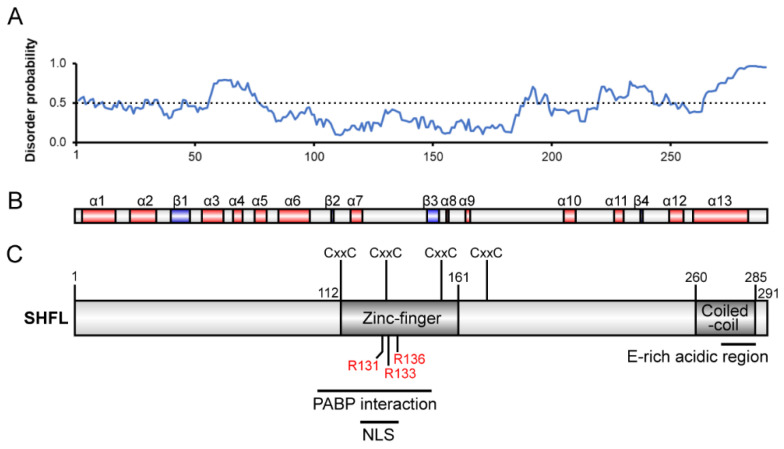
Predicted domains and features in SHFL**.** (**A**) Plot of predicted disorder probability vs. residue number (DISOPRED2 [60]), indicating a predominantly globular fold. (**B**) Prediction of secondary structural elements (PSIPRED [61]). (**C**) Annotated domain diagram of SHFL. The central CxxC motifs are highly conserved among family members (Pfam UPF0515). Other regions of interest include a putative zinc-finger domain, E-rich acidic domain, coiled-coil domain, nuclear localization sequence (NLS) and experimentally determined region necessary for interaction with PABP [47]. The locations of R131, R133 and R136 mutated in this study are indicated.

**Figure 6 viruses-13-01230-f006:**
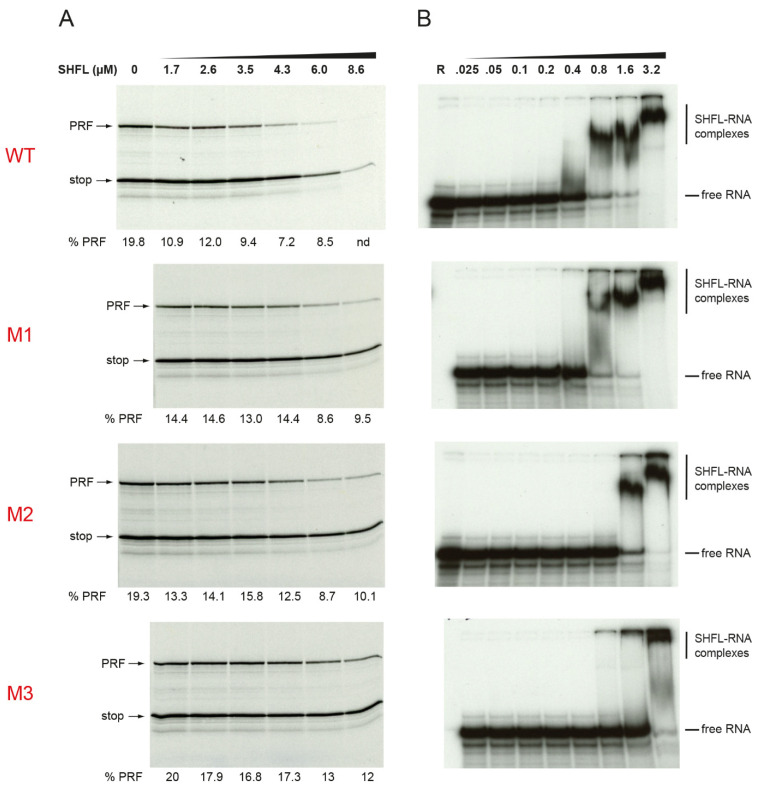
Activity of mutant derivatives of SHFL. (**A**) An mRNA derived from pDluc/SARS-CoV-2 was translated in RRL in the presence of the indicated concentrations of His-tagged SHFL (WT) or mutant derivative (M1, M2, M3). The addition of SHFL dilution buffer (DB) only is indicated. The products were resolved and analyzed as detailed in the legend to Figure 3. (**B**) EMSA analysis of binding of SHFL (WT) or mutant derivative (M1, M2, M3) to a ^32^P-labeled in vitro transcript containing the IBVmax PK. The EMSA was carried out and processed as detailed in the legend of Figure 4. The original gels from which the panels of this Figure are derived as shown in Appendix A.

**Table 1 viruses-13-01230-t001:** Sequences of recoding signals employed (recoding site underlined).

Recoding Signal	Sequence Cloned (5′-3′)
p2luc/HIV	GACAGGCTAATTTTTTAGGGAAGATCTGGCCTTCCTACAAGGGAAGGCCAGGGAATTTTCTTCAGAGCAGACCAGAG
p2luc/HIV/ifc	GACAGGCTAATTTTTTAAGGGAAGATCTGGCCTTCCTACAAGGGAAGGCCAGGGAATTTTCTTCAGAGCAGACCAGAG
p2luc/SARS-CoV	AACGTTTTTAAACGGGTTTGCGGTGTAAGTGCAGCCCGTCTTACACCGTGCGGCACAGGCACTAGTACTGATGTCGTCTACAGGGCTTTTGATATTTACAACGAAAAAGTTGCTGGTTTTGCAAAGTTCCTAAAAACTAATTGCTGTCGCTTCCAGGAGAAGGATGAGGAAG
p2luc/SARS-CoV/ifc	AACGTTTTTAAAGCGGGTTTGCGGTGTAAGTGCAGCCCGTCTTACACCGTGCGGCACAGGCACTAGTACTGATGTCGTCTACAGGGCTTTTGATATTTACAACGAAAAAGTTGCTGGTTTTGCAAAGTTCCTAAAAACTAATTGCTGTCGCTTCCAGGAGAAGGATGAGGAAG
pDluc/IBV	TAGGGCTTTAAACGGGTACGGGGTAGCAGTGAGGCTCGGCTGATACCCCTTGCTAGTGGATGTGATCCTGATGTTGTAAAGCGAGCC
pDluc/EMCV	AAGACAACGGCCGGTTTTTCAGACTCAAGGAGCGGCAGTGTCATCAATGGCTCAAACCCTACTGCCGAACGACTTGGCCAGCAAACGTATGGGATCAGCCTTTAC
pDluc/TMEV	GCAGTCGGTTTTTCAGCCATAAGGTGCGGTGCTAACCAAATCCCTAGCACCCCAGGCAGGAATTCAAAACATCCTTCTACGCCTCCTTGGCATAGAAGGCGACTG
pDluc/MuLV	CTCCCTCCTGACCCTAGATGACTAGGGAGGTCAGGGTCAGGAGCCCCCCCCTGAACCCAGGATAACCCTCAAAGTCGGGGGGCAACCCGTCACCTTCCTG
pDluc/SARS-CoV-2	CATGCTTCAGTCAGCTGATGCACAATCGTTTTTAAACGGGTTTGCGGTGTAAGTGCAGCCCGTCTTACACCGTGCGGCACAGGCACTAGTACTGATGTCGTATACAGGGCTTTTA

## Data Availability

No additional data available.

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
