# Peer review of "Modulation of Viral Programmed Ribosomal Frameshifting and Stop Codon Readthrough by the Host Restriction Factor Shiftless"

_viruses, 2021, doi:10.3390/v13071230_

Round 1

Reviewer 1 Report

This ms provides timely and significant information on a relatively new topic – the function of C19orf66. This topic is exciting as the encoded interferon-induced protein has been reported to be an inhibitor of several instances of functionally utilized programmed ribosomal frameshifting. Consequently, it has been named ‘Shiftless’, though it’s functioning is likely complex since it is also an inhibitor of several viruses whose use of programmed ribosomal frameshifting has been intensively sought with negative results. Starting with a balanced account of prior work, the Napthine et al ms. provides an important independent and thorough check of prior claims of frameshifting inhibition. It proceeds to address original questions. As to be expected from prior work of the group involved, the experimental work is carefully performed and described. The Discussion gives appropriate weight to the findings, while clearly pointing out what is needed next to bring understanding to a deeper level.

Author Response

We thank the referee for the supportive comments.

Reviewer 2 Report

In this paper, the authors used purified Shiftless recombinant protein to analyze its ability to inhibit -1PRF and translation readthrough (RT) in vitro. On one hand, this system is an artificial system, whether and to what extent the results can reflect the function and mechanism of action of Shiftless in cells are not very clear. On the other hand, some results are consistent with the previously published observations in cultured cells, suggesting that this in vitro assay could provide an alternative approach to studying the mechanisms of action of Shiftless in a system that can be easily manipulated. Based on these considerations, I think the paper can be published after some revisions, but the results need to be interpreted very carefully.

Specific comments:

  1. Figure 3. As the authors noted and explained in the Discussion, high concentration of Shiftless inhibited general translation in most cases. Then the inhibitory effect on the recoding process is hard to interpret and may just reflect that the recoding process is more sensitive to any perturbation. The authors need to show the relative translation efficiency of the stop product, as well as the PRF or RT product. This could help readers to know at what concentration Shiftless specifically inhibits the recoding process and to what extent in this in vitro assay.
  2. Figure 6. The authors showed that the three mutants displayed lower RNA-binding affinity and lower inhibitory activity, and concluded that the RNA-binding capability was required for the function of Shiftless. These mutations could also have affected the interaction of Shiftless with ribosomes. This possibility needs to be addressed, using assays in Figure 2. The results in Figure 3 that M1 did not seem to affect the translation of either the stop or the PRF product of TMEV support this possibility.

3. Minor point. The official abbreviation of Shiftless is SHFL, not SFL.

Author Response

  1. As mentioned in the manuscript, we do observe some inhibition of translation at high levels of SHFL, although this varied somewhat depending on the reporter mRNA used. However, it is clear from the gels (and the calculated % recoding efficiencies) that the inhibitory effect on recoding takes place at SHFL levels lower than those that lead to general translation inhibition. For example, in the SARS-CoV-1 translations in Figure 3, the levels of stop product remain fairly constant up to about 6.6µM added SHFL, whereas PRF is ~ 2X decreased on the first addition of SHFL (1.7 µM) and three-fold decreased by 6.6 uM SHFL. Note also that the measurement of % PRF (FS/([stop + FS] x100) normalises in principle any inhibition of translation (although we concede that effects on elongation can lead to underestimates of PRF if the frameshift product is much longer than the stop product). To address the referees concern, we now show the raw intensities of the stop and PRF products for the experiments in Figure 3 (in a new Supplementary Table). As can be seen from the raw data, with the possible exception of the HIV-1 signal (where SHFL had only a small effect on PRF) the effect on recoding is measurable and often complete before any generalised inhibitory effect on translation of SHFL occurs.

We have amended the second paragraph of the Discussion section to clarify this point and linked to the new Supplementary Table, as below:

"This inhibitory effect was observed only at concentrations of SHFL higher than those having an effect on recoding however, indicating the specificity of SHFL action in inhibiting recoding events (densitometry of the protein bands of Figure 3 is shown in the Supplementary Table). Nevertheless, subtle effects on elongation coupled with minor differences between related recoding signals and their reporter mRNA contexts might account for the observed differences in the magnitude of the SHFL effect."

  1. This is a very fair comment. Although we state in the manuscript that "Further characterisation of these mutants (M1, M2, M3) will be required to understand the molecular basis of the deficit", the possibility that these mutations affect ribosome binding has not been ruled out. The referee suggests that we use the size-exclusion chromatography approach to examine ribosome association (Figure 2), but unfortunately, this technique uses very large amounts of proteins that we do not currently have and further, we do not know whether the method will discriminate between subtle binding affinity differences. We are trying to develop a sensitive, small scale ribosome binding assay to address this as part of a planned broader mechanistic investigation of SHFL action, but these experiments are beyond the scope of the present study.

The referee correctly points out that the M1 mutant, whilst retaining measurable RNA binding activity (Figure 6) has little inhibitory activity against the TMEV frameshift signal, but some activity against the SARS-CoV-2 signal. However, these are operationally distinct signals with, in the case of TMEV, the likely necessity for competition for stem-loop binding between SHFL and 2A. It may well be that 2A is the better competitor in comparison with the M1 mutant. The lack of translation inhibition by SHFL in this experiment is likely because the highest concentration used was 3.2 µM, whereas in other experiments, noticeable inhibition is seen only at >5 µM.

  1. We apologise for this error and have corrected the text.

Round 2

Reviewer 2 Report

The paper can be published in the present form.